# A Food Insecurity Systematic Review: Experience from Malaysia

**DOI:** 10.3390/nu13030945

**Published:** 2021-03-15

**Authors:** Norhasmah Sulaiman, Heather Yeatman, Joanna Russell, Leh Shii Law

**Affiliations:** 1Department of Nutrition, Faculty of Medicine and Health Sciences, Universiti Putra Malaysia, Serdang 43400, Selangor, Malaysia; 2Malaysian Research Institute on Ageing (MyAgeing), Universiti Putra Malaysia, Serdang 43400, Selangor, Malaysia; 3Research Centre of Excellence, Nutrition and Non-Communicable Diseases (NNCD), Faculty of Medicine and Health Sciences, Universiti Putra Malaysia, Serdang 43400, Selangor, Malaysia; 4Faculty of Social Sciences, School of Health and Society, University of Wollongong, Northfields Avenue, Wollongong, NSW 2522, Australia; hyeatman@uow.edu.au (H.Y.); jrussell@uow.edu.au (J.R.); 5Department of Community Medicine and Public Health, Faculty of Medicine and Health Sciences, University Malaysia Sarawak, Kota Samarahan 94300, Sarawak, Malaysia; lslaw@unimas.my

**Keywords:** household food insecurity, Malaysia, risk factors, coping strategies, consequences

## Abstract

Living free from hunger is a basic human right. However, some communities still experience household food insecurity. This systematic literature review explored different aspects of household food insecurity in Malaysia including vulnerable groups, prevalence, risk factors, coping strategies, and the consequences of food insecurity. The review followed the Preferred Reporting Items for Systematic Reviews and Meta-Analysis (PRISMA) guidelines. Thirty-three relevant articles were selected from scientific databases such as CINAHL, Pubmed and Google Scholar, scrutiny of reference lists, and personal communication with experts in the field. The prevalence of household food insecurity in Malaysia was unexpectedly reported as high, with affected groups including *Orang Asli*, low-income household/welfare-recipient households, university students, and the elderly. Demographic risk factors and socioeconomic characteristics included larger household, living in poverty, and low education. Coping strategies were practices to increase the accessibility of food in their households. Consequences of household food insecurity included psychological, dietary (macro- and micronutrient intakes), nutritional status, and health impacts. In conclusion, this review confirmed that household food insecurity in Malaysia continues to exist. Nevertheless, extensive and active investigations are encouraged to obtain a more holistic and comprehensive picture pertaining to household food security in Malaysia.

## 1. Introduction

There are approximately 200 definitions of food security [1,2], with the most widely used definition:


*‘Food security exists when all people, at all times, have physical, social and economic access to sufficient, safe and nutritious food that meets their dietary needs and food preferences for an active and healthy life’*
[3]

The definition of food security is multidimensional, built on four ‘pillars’, namely physical availability of food, economic and physical access to food, food utilization, and stability of the three dimensions over time [4].

Food insecurity exists whenever “the availability of nutritional adequate and safe foods or ability to acquire acceptable foods in socially acceptable ways is limited or uncertain” [5]. The importance of addressing food insecurity has been recognised through the second goal of Sustainable Development Goals (SDGs) 2030 [6].

Food security has been recognized as a significant concern at the global, national, community, household and individual levels. In 2019, despite experiencing stable global food production, the FAO projected that 687.8 million people (8.9%) or one in every ten people in the world are undernourished, while 750 million people (9.7%) worldwide were experiencing severe food insecurity [7]. In several countries, particularly high-income countries, food insecurity is monitored regularly by incorporating surveys into national nutritional surveillance systems [8,9]. Food insecurity was reported as affecting approximately one in ten households (12.3%) in the U.S. [8], and 6.0% of households in Canada [9].

Another approach in studying food security has been to focus on vulnerable populations, particularly those with low socioeconomic characteristic, living in poverty and in regional or remote areas. Vulnerable populations have much higher rates of food insecurity. For example, in Puerto Rico, 40.0% of low-income households were categorized as food insecure [10]. In Alagoas, one of the poorest states in Brazil, 58.3% of households were food insecure, of which 33.1% were categorized as mild, 17.9% as moderate, and 7.3% as severe [11]. Further, in the rural area of Limpopo Province, South Africa, 53.0% of households were identified as severely household food insecure [12]. Meanwhile, a high prevalence of household food insecurity was also observed in urban households. According to a study that was conducted among rural and urban American Indian households with young children, food insecurity was reported among 44.6% rural households and 79.5% urban households [13]. Moreover, another study at Ibadan metropolis, Oyo state, Nigeria reported that 29.3% of households were food insecure [14]. The high prevalence of food insecurity globally provides solid evidence that these issues require immediate attention, particularly among disadvantaged groups.

Food insecurity has negative consequences on the health of populations, primarily due to poor dietary intakes and low dietary diversity patterns of food-insecure populations [15,16,17,18,19,20,21,22,23]. Examples of the unhealthy behaviours included high fat consumption [24], higher consumption of sugar-sweetened beverages, red/processed meat, and nuts, seeds and legumes but lower consumption of vegetables and sweet and bakery desserts [16,25], and lower intakes in marco- and micronutrients [26].

Importantly, food insecurity has been found to have consequences on nutritional status and health among children and adults. In children, mixed results were reported regarding the relationships between household food insecurity and nutritional status. Food insecurity is associated with underweight among preschool children in Antioquia, Colombia, low-income Hispanic immigrant children in Houston, the U.S., and school children in Bogotá, Colombia [19,27,28]. In wealthy countries, children experiencing persistent household food insecurity are more likely to develop obesity [29], though this is not consistently reported [30]. Significant lower physical function also has been reported among children aged 3 to 8 years in food-insecure households, and lower psychological function among children aged 12 to 17 years [31]. These impacts are present even with mild household food insecurity [32] and with children during early childhood and until middle childhood. The impacts on children in food-insecure households can be devastating, including quality of food, quantity of food, psychological aspects, and social aspects [33].

Household food insecurity also impacts the nutritional status and health of adult women. Hunger in poor countries is significantly associated with underweight [19,34], and overweight in wealthy countries [35,36]. Other aspects of health status impacted include anemia [37], hypertension, hyperlipidemia, and relative risk diabetes [38], and psychological impacts such as parental practices and depression [39]. Other groups at high risk of food insecurity include the elderly [40,41], refugees [42,43], Indigenous Peoples [44], and university students [45,46]. In addition to the adverse effects of food insecurity, affected groups also spend more for their health costs [47].

Risk factors for food insecurity include any factor that limits household resources such as money, time, information, health or even the proportion of those resources available for food acquisition. Other risk factors include those that limit employment opportunities, wage and benefit scales and social assistance benefits, or increased non-discretionary non-food expenditure such as the cost of housing and utilities, health care, taxes, childcare and the likelihood of emergencies [48]. There is also a relationship between food insecurity and poverty [49], lower per-capita income [50], lower household income, sources of income [51], households headed by females [52], chronic diseases [53], greater number of children, and lower education of mothers [54].

The physical environment also plays an important role in determining food security status. Two common factors are presence of food outlets nearby [55] and road connectivity [56,57]. For farmers or rural populations, failure in agricultural activities due to lack of arable lands [58,59] and damages caused by pests [60,61] directly influence the amount of food produced and impact their income. Food security status of the Indigenous Peoples was related to their own specific surrounding environments and lifestyles, including abandonment of traditional practices (reduced hunting of traditional food, unaffordable harvesting tools, loss of cultural hunting activities, and lack of time due to employment) and deterioration of the surrounding environment (worry over contamination of traditional foods, serious deforestation, politic instability, and contamination of water supply), and changing of lifestyles due to acculturation (loss of traditional knowledge and change in agricultural practices or higher access to western food) [62,63].

Food is a fundamental human right, as per article 25 in the Universal Declaration of Human Rights [64]. In line with the second goal of Sustainable Development Goals (SDGs) to eliminate hunger and ensure all people including poor and at-risk peoples have access to safe, nutritious and sufficient food by 2030 [6], it is expected that food insecuritywill be eliminated worldwide. Governments have an obligation not to interfere with the efforts of their citizens to attain food, provide protection when there is an infringement on rights of food, and provide opportunities to aid citizens in food attainment [65].

Malaysia is a multiethnic country located in the South East Asia region, with an area coverage total landmass of 329,847 km^2^. Recent efforts to diversify the economics of Malaysia have been successful [66], with average growth at 5.4% from 2010 to 2018. The poverty rate in Malaysia is low, at less than one percent. However, the data released were questionable as the previous Poverty Line Income (PLI) at MYR 980 (=USD 243) per household [67] was considered too low. Recently, in 2020, the PLI was revised, with an increment from MYR 980 (=USD 243) to MYR 2208 (=USD 547) per household. Under the new PLI, the poverty rate in Malaysia was expected to be at 5.6% [67]. The attention of the government is now directed to improve the quality of life among the poorest bottom 40 group (bottom 40 per cent household income group who possesses only 16.4% of the income share) who are believed to be less resilient towards economic shocks [68,69]. Based on the information that was released by the Department of Statistics Malaysia, the median income for Malaysia in 2019 was MYR 5873 (=USD 1455), MYR 3166 (=USD 784) among the bottom 40 group, MYR 7093 (=USD 1757) among the middle 40 group, and MYR 15,301 (=USD 3791) among the top 20 group [68]. Regarding ethnicity distribution in Malaysia, 67.4% were *Bumiputera*, 24.6% Chinese, 7.3% Indians, and 0.7% others [69]. *Orang Asli* was grouped under *Bumiputera* and represented 0.7% of the Malaysia population [70]. Based on data released by the Economic Planning Unit, Prime Minister’s Department, the incidence of relative poverty for *Bumiputera*, Chinese, Indians, and others in 2019 was 18.8%, 12.3%, 15.4%, and 27.9%, respectively; further, the incidence of absolute poverty for the same groups in 2019 was 7.2%, 1.4%, 4.8%, and 13.5%, respectively [71]. The data reflected that there is a huge income gap in Malaysia.

Since 1994, three series of the National Plan of Action for Nutrition for Malaysia (NPANM I to NPANM III) were established to improve the nutritional well-being of Malaysians [72,73,74] and to eradicate household food insecurity, nutritional deficiencies and malnutrition. The current NPANM III (2016–2025) has focused on food and nutrition security to promote healthy eating through increasing food quantity and quality, increasing purchasing power of food as well as reducing unhealthy eating behaviours such as skipping meals and reducing portion size [74].

In conclusion, poverty is still a major concern in Malaysia, despite excellent economic progression [67,75]. The low monthly household incomes might not sustain a stable life as prices of food and other necessities keep increasing (inflation), particularly among low-income households with single working parents and with an income of minimum wage of MYR 1100 (USD 272.65). Such observation are obvious among households in big cities such as Kuala Lumpur, Penang, Johor Bahru, Kuching, and Kota Kinabalu (urban poor) [75,76]. Attention to poverty is required due to its high association with household food insecurity in terms of purchasing power [48,49,50,51]. Similar situations are observed among single-income households, of which 89.4% of households with single working mothers were mostly categorized in the B40 group (bottom 40% of income earners). According to the Household Income and Basic Amenities Survey, the B40 group only shared 16.0% of the total income. The proportion was much lower when compared to 46.8% among the T20 group (top 20% of income earners) and 37.2% among the M40 group (middle 40% of income earners) [68]. In addition, low education levels may lead to household food insecurity, with 83.7% of the single mothers in Peninsular Malaysia having only secondary education or less [77]. Similarly, for minor ethnic groups such as Indigenous Peoples, access to education is still a concern, with high dropout rates being reported. Lack of education prevents finding a better job that can improve socioeconomic status and thus increase quality of life [78]. This might help to explain the high prevalence of household food insecurity among single mothers and Indigenous Peoples. With all the mentioned factors, Malaysia is not shielded from food insecurity, particularly at the household and individual levels. This is reflected through othe dual burden of malnutrition, micronutriens deficiencies and diet-related non-communicable diseases in Malaysia, with undernutrition remaining high [79]. The prevalence of stunting among children younger than five years old was reported to be 17.2% in 2006 and it increased to 20.7% in 2016 [79]. Micronutrient deficiencies were still a problem and the problem was given yet lower power priority when compared to overnutrition [79]. In this case, household food security becomes a topic of interest as it might be a solution to all the nutrition problems in Malaysia. Therefore, the aim of this systematic review is to provide an overall view of issues relating to household and individual food insecurity in Malaysia. Issues explored include the prevalence of household food insecurity, vulnerable groups, contributing factors of household food insecurity, coping strategies during household food insecurity, and consequences of household food insecurity. Additionally, we explored the instruments or indicators used to determine or report on household food insecurity. In this case, this systematic review is expected to answer several questions regarding food security issues among Malaysians:(a)What was the prevalence of food insecurity in Malaysia according to studies conducted there?(b)Who were the high-risk groups of household food insecurity in Malaysia?(c)What were common instruments used to assess food insecurity in Malaysia?(d)What were the common contributing factors to rather than of food insecurity among Malaysians?(e)What were the coping strategies practised by food-insecure households in Malaysia?(f)What were consequences of food insecurity reported among Malaysians?

## 2. Materials and Methods 

### 2.1. Search Strategy

This systematic literature review followed the guidelines proposed by the Preferred Reporting Items for Systematic Reviews and Meta-Analysis (PRISMA) [80]. Firstly, several worldwide electronic databases were searched, including PubMed and CINAHL [81]. The descriptors for the search were ‘food insecurity’ or ‘food security’ AND ‘Malaysia’. Secondly, to maximize the number of available articles, a search was also carried out on Google Scholar using similar descriptors. Thirdly, a researcher scrutinized the reference sections of all relevant articles retrieved to detect any further published articles not identified during the database searches [81]. Fourthly, journal articles were obtained through personal communication with experts in the field [82]. Two authors screened all titles and abstracts of the selected articles in order to filter out irrelevant and duplicated articles.

### 2.2. Selection Criteria

One independent researcher (N.S.) read the full articles. Inclusion criteria included (a) original/primary peer-reviewed journal articles; (b) the terms ‘food insecurity’ or ‘food security’ were clearly defined; (c) household and individual food insecurity were prioritized; (d) indicators to measure food insecurity were clearly defined; (e) the study was conducted in Malaysia; (f) written in the English or Malay languages; (g) conducted by local or foreign researchers; and (h) articles that were published from January 2000 to December 2020. No limitation was set for target population or age groups. With all the criteria applied, 46 articles qualified for review. The chosen articles were reviewed by a second independent researcher (L.L.S.). The final decision on the selection of articles was made based on the consensus of the researchers through discussion. In the process, two full articles were excluded as the two articles focused on the national and community levels of food insecurity, that is the availability dimension, which violated inclusion criterion (c).

### 2.3. Data Extraction

Information from each of the selected articles was extracted by an independent researcher (N.S.). Another member of the research team (L.L.S.) screened the information in order to clarify that the information extracted was accurate. Firstly, the articles were categorized based on target groups of the studies and then details were extracted relating to the publications (authors and year of publication) and the studies (study designs, respondents, study location, and instruments used to measure household food security). The studies were divided into four components based on the scope of the studies, namely prevalence, contributing factors of household food insecurity, coping strategies during household and individual food insecurity, and consequences of household food insecurity.

### 2.4. Risk of Bias Assessment

The quality of the articles was assessed by two authors (N.S. and L.L.S.) using the Joanna Briggs Institute Critical Appraisal Tool for use in JBI systematic reviews [83]. Scores were given based on the consensus that was reached between two authors. The quality of the articles was assessed against quantitative and qualitative checklists, respectively [84,85]. The quality of cross-sectional studies was based on the level of detail provided for inclusion criteria, descriptions of the respondents, assessments of exposures and outcomes, elaboration on confounding variables, and statistical analysis (Table 1). 

The quality of studies utilising qualitative research was assessed against a number of criteria, including congruities between the philosophical perspective and methodology; methodology and research objectives; methodology and methods for collecting data; methodology and analysis of data; methodology and interpretation of results; presence of a statement to locate the researcher culturally or theoretically; the influence of the researcher on research; representative of participants and their voices; information on ethical approval; and construction of the conclusion based on interpretation of data needed to be elaborated clearly (Table 2).

## 3. Results

The systematic review included 46 articles after removal of the irrelevant and duplicated articles. Figure 1 illustrates the selection processes.

For quantitative studies, 17 articles (40.5%) met six and more items (high quality) in the risk of bias checklist for analytical cross-sectional studies, while another 18 articles (42.9%) fulfilled three to five items (fair quality). Finally, seven articles (16.7%) met less than three items. The studies that met the criteria of the checklist for analytical cross-sectional studies are shown in Appendix A. Meanwhile, for qualitative studies, two articles met eight items in the risk of bias checklist for qualitative research, while one article met seven items and another met five items. The studies that met with the criteria of the checklist for qualitative research are shown in Appendix A.

Generally, four different scales were used to measure household food insecurity during quantitative research, namely the United States Department of Agriculture Household Food Security Survey Module (HFSSM) [86], the Radimer/Cornell Hunger and Food Insecurity Instrument [87], the Malaysian Coping Strategy Instrument (MCSI) [88], and the Household Food Insecurity Access Scale (HFIAS) [89]. Among them, the Radimer/Cornell Hunger and Food Insecurity Instrument was the most widely used instrument as it was used in 21 of the total publications. It was followed by the HFSSM in 12 publications. Meanwhile, the MCSI appeared in three publications. Other instruments included the Household Food Insecurity Access Scale (HFIAS) and the Food Security Tool for Elderly, which were used in two and one publications, respectively.

### 3.1. The Prevalence of Food Insecurity 

Five main Malaysian subpopulations, namely Indigenous People, adolescents/adults/low-income households/welfare-recipient households, university students, elderly population, and migrant workers were identified. The *Orang Asli* (Indigenous People) had rates of food insecurity from 81.2 to 88.0% [90,91,92,93]. Adolescents/adults/low-income households/welfare-recipient households experienced a prevalence of food insecurity ranging from 47.2% to 100.0% [54,88,94,95,96,97,98,99,100,101,102,103]. Food insecurity in university students ranged from 22.0 to 70.0% [104,105,106,107,108,109,110]. The prevalence of household food insecurity among older people ranged from 6.9% to 27.7% [108,111,112,113,114], as shown in Table 3. Lastly, one study on migrant workers reported that 5.6% of migrants were food insecure [115].

For *Orang Asli*, the prevalence rates that were measured using Radimer/Cornell Hunger and Food Insecurity Instruments were between 81.2% and 88.0%. The prevalence of adolescents/adults/low-income households/welfare recipients was in the range of 50.6%–100.0% measured using the Radimer/Cornell Hunger and Food Insecurity Instruments, 68.1%-–73.8% using the MCSI, and 47.2%–64.3% using the HFIAS. The prevalence of food insecurity among university students was between 43.5% and 69.4% when measured using the HFSSM and 22.0% when using the Radimer/Cornell Hunger and Food Insecurity Instruments. Lastly, the prevalence of elderly food insecurity was 10.4% and 22.9% measured by the HFSSM, 6.9% by the Radimer/Cornell Hunger and Food Insecurity Instruments, 27.7% by the Food Security Tool for Elderly, and 5.6% by the HFIAS.

### 3.2. Contributing Factors to Food Insecurity 

Most studies (30/32 articles) reported on socioeconomic factors contributing to household food insecurity (Table 4). Factors included poverty, low food expenditure, low education, and depletion of assets. University students experienced financial hardship due to inadequacy of student loans/scholarships. Limited income, increased expenditure on tuition fees and cost of living were primary contributing factors for university students. The structure of the household was a key contributing factor to food insecurity (19/32), including female-headed households, larger household, more children at school age, living in a rural residential area, and disabled household member(s). In addition, Siti Farhana et al. [108] reported that mental health was a contributing factor to food insecurity among the elderly. Law et al. [117] found that constraints in practicing traditional lifestyles among *Orang Asli* was associated with food insecurity. These constraints included failure in agriculture, ineffectiveness in traditional food-seeking activities, weather, and water issues. 

### 3.3. Coping Strategies 

Coping strategies were divided into two main categories, namely food-related and non-food-related coping strategies. Food-related coping strategies included reducing the amounts of food consumed, borrowing money to buy food, borrowing food, and skipping meals [93,117]. Non-food-related coping strategies related to improving participants’ socioeconomic backgrounds by undertaking odd jobs, borrowing money from relatives, and delaying payment of bills [93,117,118,131,133], as shown in the Table 4.

### 3.4. Consequences of Food Insecurity 

The consequences of food insecurity across the four high-risk populations were explored by 21 articles. Children experienced unhealthy body weight status (low weight for age and low height for age) and inadequate dietary intakes (low diet quality and insufficient intake of nutrients) [117]. Women experienced inadequate dietary intakes such as low intake of energy, vitamins, fat and sodium, as well as low diet quality, and low intake of certain food groups (meat, fish, poultry, and legumes) [88], or high fat and sodium [90]. Household food insecurity contributed to poor health through higher body mass index (BMI), overweight and obesity, abdominal obesity, elevation in plasma glucose, cholesterol, low-density lipoprotein cholesterol and metabolic syndrome, and low quality of life, as shown in Table 4.

Food insecurity in university students triggered anxiety and poor physical health through lack of energy, skipping classes, as well as higher BMI and triglyceride levels [130]. In elderly people, household food insecurity was linked to unhealthy body status (lower height) and inadequate dietary intake (low fat) [111], or higher intake of fat, oil, sugar, and salt [112].

## 4. Discussion

In this case, research on food insecurity at the household/individual level in Malaysia started late with the first article published in 2001 [119]. Since then, food insecurity issues have continued to gain attention at a rapid pace, as shown in this review. This review provides critical holistic information about food insecurity in high-risk populations in Malaysia, including prevalence, contributing factors, coping strategies, and consequences of household food insecurity. Most of the articles published reported on small-scale research projects focusing on high-risk households, particularly low-income households at the provincial level. During the early period covered by this review, limited attention from researchers was given to household food insecurity as Malaysia was experiencing rapid economic development, particularly in the 1980s and 1990s [134]. In 2014, the Malaysian Adult Nutrition Survey, a nationally representative study, included food insecurity indicators (skipping breakfast and relying on cheap food due to financial constraints) in the study [135]. The use of only two items does not provide a comprehensive picture of food security in the Malaysian population. Improvement is a necessity. 

This review identified two groups experiencing high levels of food insecurity, *Orang Asli* women and low-income households/welfare-recipient households. The 10th Malaysian Plan (2011–2015) recognized *Orang Asli* as a disadvantaged group, with limited access to cchild education, water supply, electricity, and health care services [136]. Approximately 31.2% of the *Orang Asli* population (0.7% of Malaysia population) were living below PLI in 2010 and approximately 20.0% experienced extreme poverty [137]. Poor and extremely poor households, single parents, and family members with disabilities are eligible to receive financial and food assistance from government agencies. According to the high prevalence of household food insecurity among *Orang Asli* and low-income households/welfare-recipient households, it can be concluded that food aid and financial aid provided to the needy households brought only a short-term relief but did not solve the root problem (poverty). Therefore, there is a high probability that welfare recipients remained poor and were unable to achieve self-sustainability, becoming dependent on the aid received. Examples of studies that involved low-income households in other countries are Loopstra and Tarasuk in Canada [138], Weigel, Armijos, Racines, Cevallos, and Castro in Ecuador [139], and and Miller [140] in the U.S. Further, studies on Indigenous Peoples were conducted by Jasmin, Martin, Yanling, and Piort [141] in Canada, Willows, Veugelers, Raine, and Kuhle [142] in Canada, and Temple and Russell [143] in Australia.Certain characteristics of these groups have been recognized as resulting in higher vulnerability and less resilience toward household food insecurity and extra attention is required in order to address these issues.

The range of findings on the prevelance household food insecurity can be explained by differences in the instruments used to assess food insecurity. These include the United State Department of Agriculture HFSSM, the Radimer/Cornell Hunger and Food Insecurity instrument, and the HFIAS. These tools were developed in high-income countries and were not adapted to the Malaysian culture, thus further validity and reliability studies are required to ensure that these tools are able to capture the true levels of food insecurity in Malaysia [144]. 

University students were found to be at high risk of food insecurity but the reported prevelance varied from 43.5% [130] to 67.7% [128,129]. The variations in the prevalence of food insecurity in these studies may be due to different study locations and numbers of respondents. The study by Nur Atiqah et al. [130] was conducted involving 124 undergraduate students at one public university located at Klang Valley, while the studies reported by Norhasmah et al. [128] and Siti Marhana et al. [129] involved 484 undergraduate students at four public universities in Peninsular Malaysia. Other countries have also reported variations in the prevalence of food insecurity among university students, ranging from 35.3% to 48.0% [145,146,147]. 

Lastly, elderly people are another group for concern in terms of increased household food insecurity although evidence from Malaysia is still limited, with only four articles published since 2017. Further research in this population is essential in Malaysia as the population is ageing rapidly, with 15% of the population expected to be aged over 65 years by 2030 [148]. Evidence from the US found that 13.6% of seniors were marginally food insecure, 7.7% were food insecure, and 2.9% were very low food secure [149]. Other studies conducted in various countries such as Australia [150], Turkey [151] and Brazil [41] have reported that the prevalence of food insecurity among elderly ranges from 13.0% to 21.8%. 

In this systematic review, low socioeconomic background was identified as a key contributing factor to household food insecurity in all high-risk groups. These findings were consistent with studies from other countries among Indigenous Peoples [152,153], low-income households/welfare-recipient households [138,154] and elderly people [155,156]. Households with a higher income showed high-quality diets as well as more resilience towards any shock that might impact accessing food [157]. Another common contributing factor to household food insecurity was household structure including a larger household, more children, and female-headed households. Household structure affected food utilization among household members. A large household indicated that household members had to share limited amounts of food [158].

A lack of evidence was found relating to health status as a contributing factor to food insecurity in Malaysia compared to food insecurity among elderly people in other countries. Fernandes et al. [155] reported greater odds of having a chronic disease, reduced regular medical visits, and ceasing medication use due to financial issues among food-insecure elderly people in Portugal. Similarly, elderly respondents in the United States with multiple chronic conditions (two or more conditions) had a higher risk of being food insecure compared to elderly respondents with 0 to one condition. Older adults managing chronic diseases have greater economic constraints that impact their ability to acquire sufficient money to buy food [159]. 

In Indigenous populations, the ability to access traditional food from their surroundings was a unique characteristic and challenges to practicing their traditional food-seeking activities were considered to contribute to a higher prevalence of food insecurity. As such, it was not only economic access but physical access to food that increased risk of food insecurity. Consistent factors relating to food insecurity such as reduced hunting of traditional foods, high cost of harvesting tools, loss of cultural hunting activities, lack of time due to employment, and worry over contamination of traditional foods were reported by Schuster et al. [63] among Indigenous Peoples in Canada. 

Two types of coping strategies were identified to manage household food insecurity. Food-related coping strategies involved activities pertaining to short-term food acquisition activities for overcoming food shortage. Meanwhile, non-food-related coping strategies or long-term livelihood coping strategies included improving the financial status of households so that more resources were available to buy more food [160]. Other coping strategies included the acceptance of or resignation to the food shortage situation. Lastly, emotion-focused coping strategies were linked to stress without having a solution [161]. Examples of emotion-focused coping included aspirational (having faith that good fortune will prevail), resignation (isolation from receiving help), distraction (mind tricks), and anger/violence (frustration). Such behaviours were practiced consistently by food-insecure household globally, for example as reported by in Kisi et al. [162] from Ethiopia, Ghimire [163] from Nepal, Rukundo, Oshaug, Andreassen, and Kikafunda [164] from Uganda, and Mabuza, Ortmann, and Wale [165] from Swaziland. 

In Malaysia, household food insecurity has been found to compromise dietary intakes of respondents in terms of diet quality, intakes of certain food groups, and intakes of macro- and micronutrients among low-income households, university students, and elderly people in Korea, Ghana and India [17,166,167,168]. Reduced diet quality among food-insecure respondents was due to the high cost of quality food and financial constraints that inhibited them from making better food choices. As such, their choice of food shifted to low-quality food that were cheaper and highly accessible [44]. 

In addition to the negative influences of dietary constraints, psychological issues resulting from food insecurity were only reported amongst university students [133]. However, in other countries, there are reports of a relationship between food insecurity and psychological problems such as increased risk of developing mental disorders within households that included children [169,170]. The vulnerability of children towards developing persistent depression/anxiety and hyperactivity/inattention could be explained through the association of household food insecurity with poverty, marital discord, single parenthood, violence, parental substance abuse, and psychopathy, which influence child development and mental health [171].

Household food insecurity in Malaysia was found to be closely related to poor nutritional status, either underweight or overweight among women, children, university students, and elderly people. Relationships between adult underweight and household food insecurity have also been reported elsewhere, whereby adults compromise their own diet as a coping strategy to ensure healthy growth among children [19,34]. Overseas countries have also reported that household food insecurity was associated with overweight and obesity [36,172]. Overweight and obesity in food insecurity may result from greater consumption of high energy dense foods which are more accessible and cheaper to buy [173]. 

### 4.1. Limitations of This Study

There are several limitations to this systematic review. Generally, the limitations relate to the following aspects: article selection, characteristics of the articles, methodology, and generalization. Regarding article selection, this systematic review involved only peer-reviewed journal articles, without considering other forms of knowledge that might exist in reports, proceedings, booklets, or even posters. However, the latter documents are mainly unpublished and difficult to track. In addition, although four procedures were used to search for relevant journal articles, only two scientific databases were searched, which may have reduced access to some relevant articles and this is a key limitation. The two electronic databases were selected as the university subscribed to these two databases, providing access to full text of the articles. Limitations also relate to the methodology of the studies included in this review. The majority of the articles used a cross-sectional study design in quantitative research and a case study design in qualitative research. As such, it is not possible to determine cause and effect inferences of risk factors and consequences of household food insecurity. Nonetheless, more aspects of food security can be explored by applying new methods, particularly environmental factors under contributing factors. For example, the geographical information system (GIS) can be used to identify high-risk locations within a district. Multidimensional and comprehensive approaches provide an overview of food security, but current studies mostly focus on access and utilization aspects, also known as household and individual food security. A more holistic view can be observed by applying the Food Insecurity Multidimensional Index, which can capture the four dimensions of food security, namely availability, access, utilization, and stability. Notably, the use of different food insecurity measurement tools does not allow for accurate comparison across studies. Another limitation is that the instruments used in most of the studies were not well validated for use in the Malaysian context as they were developed in other countries. Generalization of the reviewed articles is also a limitation of this review. The sample size of all the studies were small and generalization of the findings to large population is not possible. Additionally, the target population focused only on high-risk groups such as low-income households, university students, Indigenous Peoples, and elderly. Future work could consider a national survey in order to gain a deeper general understanding towards food security in Malaysia and could include several groups such as Indigenous groups at Sabah and Sarawak as well as refugees. Despite these limitations, the results of this study are still useful for health practitioners, nutritionists, non-governmental organizations, and policy makers in planning strategies to reduce the burdens of household food insecurity among high-risk groups.

### 4.2. Recommendation

The studies provide clear evidence of household food insecurity among certain high-risk groups in Malaysia. The government must enforce the National Nutrition Policy of Malaysia 2005 as an effort to eradicate household food insecurity. Firstly, under the policy, a systematic surveillance system to monitor the progression of household food insecurity in Malaysia on a regular basis (for a specific period of five years) could be established. In addition, more target groups, such as individuals with HIV [174], people living with disabilities [175], immigrants [176], and refugees [43], should be considered in surveillance systems, as from the previous studies from foreign countries, they are considered as high-risk groups and require more attention compared to general population. Moreover, it is recommended that a stronger study design such as cohort study design be used in order to determine the cause and effect of household food insecurity. Secondly, through improved understanding of household food insecurity in Malaysia, strategies and interventions can also be implemented to address household food insecurity among high-risk groups. Interventions and planning should include improvements in household income, access to education, and employability. Lastly, through this systematic review, more research activities pertaining to household food insecurity should be encouraged in order to improve understanding of the undesired issues. Through this systematic review, several limitations are detected. The research gaps should be addressed by the researchers while planning for future research.

## 5. Conclusions

Having enough food to eat is a basic human right yet household food insecurity exists globally. Malaysia is moving forward with several research activities aimed to improve understanding of the problems. Through analysis on the studies, objectives of this systematic review were achieved, with information obtained on the prevalence of household food insecurity, risk factors for household food insecurity, coping strategies during household food insecurity, and consequences of household food insecurity. Risk factors, coping strategies, and consequences are important indicators that can be used to determine high-risk groups and even levels of food security status. The findings revealed that food security in Malaysia seems to be similar to that in other countries. Nevertheless, it is important to acknowledge that the core of the problem might be different from country to country. Whatever solution is working in other countries might not be effective in Malaysia. Therefore, this systematic review helps in understanding household food insecurity in a Malaysian context. With the available information, context-specific intervention programmes can be implemented to tackle household food insecurity among vulnerable groups in Malaysia. The information is important to act as a reminder that the stream of rapid economic expansion may not be beneficial to every single community in the nation. There are always certain vulnerable groups that cannot follow the pace and fall behind. In this matter, a policy that aims to help these vulnerable groups in terms of improving their socioeconomic background can be recommended and thus securing food supply and intake. This can be achieved through the unity of Malaysians by establishing a conducive and harmonious environment that encourages learning and helps to improve employment rates and increase the mean monthly household income.

## Figures and Tables

**Figure 1 nutrients-13-00945-f001:**
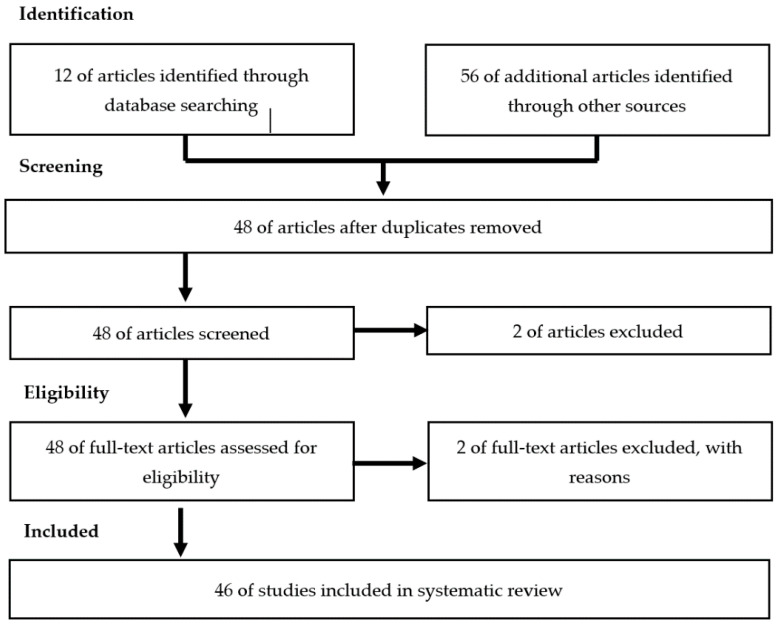
Article Selection Processes.

**Table 1 nutrients-13-00945-t001:** Checklist for analytical cross-sectional studies in JBI Critical Appraisal Tool [84].

No.	Checklist
1	Were the criteria for inclusion in the sample clearly defined?
2	Were the study subjects and the setting described in detail?
3	Was the exposure measured in a valid and reliable way?
4	Were objective, standard criteria used for measurement of the condition?
5	Were confounding factors identified?
6	Were strategies to deal with confounding factors stated?
7	Were the outcomes measured in a valid and reliable way?
8	Was appropriate statistical analysis used?

**Table 2 nutrients-13-00945-t002:** Checklist for qualitative research in JBI Critical Appraisal Tool [85].

No.	Checklist
1	Is there congruity between the stated philosophical perspective and the research methodology?
2	Is there congruity between the research methodology and the research question or objectives?
3	Is there congruity between the research methodology and the methods used to collect data?
4	Is there congruity between the research methodology and the representation and analysis of data?
5	Is there congruity between the research methodology and the interpretation of results?
6	Is there a statement locating the researcher culturally or theoretically?
7	Is the influence of the researcher on the research, and vice- versa, addressed?
8	Are participants, and their voices, adequately represented?
9	Is the research ethical according to current criteria or, for recent studies, and is there evidence of ethical approval by an appropriate body?
10	Do the conclusions drawn in the research report flow from the analysis, or interpretation, of the data?

**Table 3 nutrients-13-00945-t003:** Prevalence of and groups at risk of food insecurity in Malaysia.

No.	Authors	Year	Study Design	Respondents (Target Groups)	Study Location	Instrument	Prevalence	Quality Assessment
Orang Asli (Indigenous Peoples)								
1.	Zalilah, M.S. and Tham, B.L. [116]	2002	Cross-sectional study	*n* = 64 Women Orang Asli (Indigenous Peoples) (information on household food insecurity) and child (weight and height)	Hulu Langat District of Selangor, State	Radimer/Cornell Hunger and Food Insecurity Instrument	81.2% food insecurity (20.3% household food insecurity 32.8% individual food insecurity 28.1% child hunger)	2/8
2.	Nurfahilin, T. and Norhasmah, S. [92]	2015	Cross-sectional study	*n* = 92 Women Orang Asli	Gombak District of Selangor, State	Radimer/Cornell Hunger and Food Insecurity Instrument	88% food insecurity (48.9% household food insecurity 21.7% individuals food insecurity 17.4% child hunger)	4/8
3.	Chong, S.P., Geeta, A. and Norhasmah, S. [90]	2018	Cross-sectional study	*n* = 222 Orang Asli (Indigenous) Women Non-pregnant and non-lactating Non-vegetarian No chronic diseases No changes in food habits in past six months	Kuala Langat District of Selangor, State	Radimer/Cornell Hunger and Food Insecurity Instrument	82.9% food insecurity (29.3% household food insecurity, 23.4% individual food insecurity, 30.2% child hunger)	7/8
4.	Chong, S.P., Geeta, A. and Norhasmah, S. [91]	2019	Cross-sectional study	*n* = 222 Orang Asli (Indigenous) Women Non-pregnant and non-lactating Non-vegetarian No chronic diseases No changes in food habits in past six months to reduce weight	Kuala Langat District of Selangor, State	Radimer/Cornell Hunger and Food Insecurity Instrument	82.9% food insecurity (29.3% household food insecurity, 23.4% individual food insecurity, 30.2% child hunger)	8/8
5.	Law, L.S., Norhasmah, S., Gan, W.Y. and Mohd Nasir, M.T. [117]	2018a	Case study (qualitative study)	In-depth interviews 61 women (Senoi, Proto-Malay and Negrito) FGD 19 women proto Malay	Perak, Selangor, and Pahang, Malaysia	-	-	8/10
6.	Law, L.S., Norhasmah, S., Gan, W.Y., Siti’Asyura, A., and Mohd Nasir, M.T. [118]	2018b	Case study (qualitative study)	*n* = 61 *Orang Asli* women Food aid-recipient household Non-pregnant and non-lactating	Perak, Selangor, and Pahang, Malaysia	-	-	8/10
Low-income households/welfare-recipient households								
7.	Zalilah, M.S. and Ang, M. [119]	2001	Cross-sectional study	*n* = 137 Women (information on household food insecurity) and child (weight and height)	Kuala Lumpur	Radimer/Cornell Hunger and Food Insecurity Instrument	65.7% food insecurity (27.7% household food insecurity 10.9% individual food insecurity 27.0% child hunger)	5/8
8.	Zalilah, M.S. and Khor, G.L. [100]	2004	Cross-sectional study	*n* = 200 Women of poor households in a rural community in Malaysia	Sabak Bernam District of Selangor state	Radimer/Cornell Hunger and Food Insecurity Instrument	58.0% food insecurity (14% household food insecurity 9.5% individual food insecurity 34.5% child hunger)	7/8
9.	Zalilah, M.S. and Khor, G.L. [101]	2005	Cross-sectional study	*n* = 200 Women of poor households in a rural community in Malaysia	Sabak Bernam District of Selangor State	Radimer/Cornell Hunger and Food Insecurity Instrument	58.0% food insecurity (14.0% household food insecurity 9.5% individual food insecurity 34.5% child hunger)	6/8
10.	Zalilah, M.S. and Khor, G.L. [102]	2008	Cross-sectional study	*n* = 200 Women of poor households in a rural community in Malaysia	Sabak Bernam District of Selangor State	Radimer/Cornell Hunger and Food Insecurity Instrument	58.0% food insecurity (14.0% household food insecurity 9.5% individual food insecurity 34.5% child hunger)	7/8
11.	Norhasmah, S., Zalilah, M.S., Mirnalini, K., Mohd Nasir, M.T and Asnarulkhadi, A.S. [88]	2011	Cross-sectional study	*n* = 301 Women Non-pregnant and non-lactating, living in rural and urban areas	Tumpat and Kota Bharu Districts of Kelantan, State	Malaysian Coping Strategy Instrument (MCSI)	68.1% food insecurity (33.9% moderately food insecurity, 34.2% severely food insecurity)	6/8
12.	Norhasmah, S., Zalilah, M.S. and Rohana, A.J [99]	2012a	Cross-sectional study	*n* = 301 Women Non-pregnant and non-lactating, living in rural and urban areas	Tumpat and Kota Bharu Districts of Kelantan, State	Malaysian Coping Strategy Instrument (MCSI)	68.1% food insecurity (33.9% moderately food insecurity, 34.2% severely food insecurity)	6/8
13.	Mohamadpour, M., Zalilah M.S., Keysami M.A. [120]	2012	Cross-sectional study	*n* = 169 Palm plantation women Non-pregnant and non-lactating	Nilai, Negeri Sembilan State	Radimer/Cornell Hunger and Food Insecurity Instrument	85.2% food insecurity (24.9% household food insecurity, 19.5% individual food insecurity 40.8% child hunger)	6/8
14.	Nik Aida Adibah, N.A.A. and Norhasmah S. [109]	2013	Cross-sectional study	*n* = 80 Women	Kampung Pulau Serai, Dungun District of Terengganu, State	Radimer/Cornell Hunger and Food Insecurity instrument	93.75% food insecurity (51.25% household food insecurity 23.75% individuals food insecurity 18.75% child hunger)	3/8
15.	Zalilah, M.S., Norhasmah, S., Rohana, A.J., Wong, C.Y, Yong, H.Y., Mohd Nasir, M.T., Mirnalini, K., and Khor, G.L. [121]	2014	Cross-sectional study	*n* = 625 Women Low-income communities	3 states of Peninsular Malaysia (Selangor, Negeri Sembilan and Kelantan)	Radimer/Cornell Hunger and Food Insecurity Instrument	78.4% food insecurity (26.7% household food insecurity 25.3% individual food insecurity 26.4% child hunger)	7/8
16.	Alam, M.D., Siwar, C., Wahid, A.N.M. and Abdul Talib, B. [122]	2016	Cross-sectional study	*n* = 460 Households from urban and rural areas East Coast Economic Region (ECER) (e-kasih database)	Kelantan, Terengganu Pahang States	United States Agency for International Development Household Food Insecurity Access	47.2% food insecurity (23.3% mildly food insecurity 14.3% moderately food insecurity 9.6% are severely food insecurity)	1/8
17.	Yong, P.P. and Norhasmah, S. [110]	2016	Cross-sectional study	*n* = 109 Chinese women aged between 30 and 55 years old	Program Perumahan Rakyat (PPR) Kuala Lumpur	Radimer/Cornell Hunger and Food Insecurity Instrument	50.6% food insecurity (18.5% household insecurity 14.8% individual insecurity 17.3% child hunger)	1/8
18.	Norhasmah, S., Zalilah, M.S., Mirnalini, K., Mohd Nasir, M.T. and Asnarulkhadi, A.S. [123]	2012b	Cross-sectional study	*n* = 103 Women Non-pregnant and non-lactating welfare-recipient households	Hulu Langat District of Selangor, State	Malaysian Coping Strategy Instrument (MCSI)	73.8% food insecurity (39.8% moderate food insecurity, 34.0% severely food security)	4/8
19.	Siti Marhana, A.B. and Norhasmah, S. [124]	2012	Cross-sectional Study	*n* = 80 Women and men Zakat recipients	Bukit Martajam, Pulau Pinang, State	Radimer/Cornell Hunger and Food Insecurity Instrument	100.0% food insecurity (5.0% household food insecurity, 30.0% individual food insecurity 65.0% child hunger)	3/8
20.	Ihab, A.N., Rohana, A.J., Wan Manan, W.M., Wan Suriati, W.N., Zalilah, M.S., and Mohamed Rusli, A. [54]	2012a	Cross-sectional study	*n* = 223 Women Non-pregnant and non-lactating welfare-recipient households	Bachok District of Kelantan, State	Radimer/Cornell Hunger and Food Insecurity instrument	83.9% food insecurity (29.6% household food insecurity 19.3% individuals food insecurity 35.0% child hunger)	6/8
21.	Ihab, A.N., Rohana, A.J., Wan Manan, W.M., Wan Suriati, W.N., Zalilah, M.S., and Mohamed Rusli, A. [103]	2012b	Cross-sectional study	*n* = 223 Women Non-pregnant and non-lactating welfare-recipient households	Bachok District of Kelantan, State	Radimer/Cornell Hunger and Food Insecurity instrument (PJN)	83.9% food insecurity (29.6% household food insecurity 19.3% individuals food insecurity 35.0% child hunger)	6/8
22.	Ihab, A.N., Rohana, A.J., Wan Manan, W.M., Wan Suriati, W.N., Zalilah, M.S., and Mohamed Rusli, A. [94]	2012c	Cross-sectional study	*n* = 223 Women Non-pregnant and non-lactating welfare-recipient households	Bachok District of Kelantan State	Radimer/Cornell Hunger and Food Insecurity instrument	83.9% food insecurity (29.6% household food insecurity 19.3% individuals food insecurity 35.0% child hunger)	7/8
23.	Ihab, A.N., Rohana, A.J., Wan Manan, W.M., Wan Suriati, W.N., Zalilah, M.S., and Mohamed Rusli, A. [95]	2013	Cross-sectional study	*n* = 223 Women Non-pregnant and non-lactating welfare-recipient households	Bachok District of, Kelantan State	Radimer/Cornell Hunger and Food Insecurity Instrument	83.9% food insecurity (29.6% household food insecurity 19.3% individuals food insecurity 35.0% child hunger)	7/8
24.	Cooper, E.E. [125]	2013	Cross-sectional study	*n* = 70 Households Children (*n* = 104) aged between 6 months and 6 years old Malay fishing villages	Sematan subdistrict of Lundu at the southwest coast of Sarawak	Household Food Insecurity Access Scale (HFIAS)	64.3% food insecurity (10.0% mildly food insecurity 25.7% moderately food insecurity 28.6% severely food security)	2/8
25.	Roselawati, M.Y., Wan Azdie, M.A.B., Aflah, A., Jamalludin, R., and Zalilah, M.S. [126]	2017	Cross-sectional study	*n* = 128 Malay women with their children Age 21 to 59 years olds	Kuantan District of Pahang State	Radimer/Cornell Hunger and Food Insecurity instrument	77.0% food insecurity (52.0% house old food insecurity, 9.0% individual food insecurity 16.0% child hunger)	7/8
26.	Mohamad Hasnan, A., Rusidah, S., Ruhaya, S., Nur Liana, A.M., Ahmad Ali, Z., Wan Azdie, A.B., and Tahir, A. [96]	2020	Cross-sectional study	*n* = 2962 Adults	National representative sample in Malaysia	Six items adapted from USDA instruments (United States Department of Agriculture)	Quantity insufficiency: 25.0% Variety insufficiency: 25.5% Reduced size of meal: 21.9% Skipped meal: 15.2% Rely on cheap food and affordable food: 23.7% Could not afford to feed their children with various food: 20.8%	4/8
27.	Nor Syaza Sofiah, A.; Norhasmah, S. [97]	2020	Cross-sectional study	*n* = 140 Adults	Two national primary schools	Radimer/Cornell Hunger and Food Insecurity instrument	80.7% food insecurity (26.4% household food insecurity, 27.9% individual food insecurity, 26.4% child hunger)	4/8
28.	Noratikah, Norhasmah and Siti Farhana [98]	2019	Cross-sectional study	*n* = 129 Mothers aged 20 to 59 years	Low household income	Radimer/Cornell Hunger and Food Insecurity instrument	55.8% food insecurity (7.0% household food insecurity, 30.2% individual food insecurity, 18.6% child hunger)	7/8
29.	Susanti, A., Norhasmah, S., Fadilah, M.N., and Siti Farhana, M. [127]	2019	Cross-sectional study	*n* = 160 Households that comprised pairs of mothers and children aged 13–17 years	Mentakab, Temerloh district, Pahang	Radimer/Cornell Hunger and Food Insecurity instrument	48.8% food insecurity (20.0% household food insecurity, 13.8% individual food insecurity, 15.0% child hunger)	8/8
30.	Norhasmah, S.; Zalilah, M.S.; Mohd Nasir, M.T.; Asnarulkhadi, A.S. [93]	2010	Case study (qualitative study)	*n* = 57 20–50-year-old women	Selangor and Negeri Sembilan, Malaysia	-	-	7/10
University students								
31.	Norhasmah, S., Zuroni, M.J. and Siti Marhana, A.B. [128]	2013	Cross-sectional study	*n* = 484 Undergraduate Public University students	Peninsular Malaysia	USDA instruments (United States Department of Agriculture)	67.1% food insecurity (44.4% low food insecurity, 22.7% very low food insecurity)	1/8
32.	Siti Marhana, A.B., Norhasmah, S. and Husniyah, A.R [129]	2014	Cross-sectional study	484 Public Undergraduate Public University students	Peninsular Malaysia	USDA instruments (United States Department of Agriculture)	67.1% food insecurity (44.4% low food insecurity, 22.7% very low food insecurity)	2/8
33.	Nur Atiqah, A., Norazmir, M.N, Khairil Anuar, M.I., Mohd Fahmi, M. and Norazlanshah, H. [130]	2015	Cross-sectional study	*n* = 124 University students	Universiti Teknologi MARA (UiTM) Puncak Alam Selangor, State	Adults Food Security Survey Module (AFSSM)	43.5% food insecurity	5/8
34.	Izwan Syafiq, R.; Asma, A.; Nurzalinda, Z.; Rahijan, A.W.; Siti Nur Afifah, J. [104]	2019	Cross-sectional study	*n* = 96 University students	Two selected public universities in Terengganu	Radimer/Cornell Hunger and Food Insecurity instrument	22% food insecurity (14% low food security, 8% very low food security)	1/8
35.	Khairil, A.; Noralanshah, H.; Farah Syafeera, I.; Nazrul, H.I.; Muhammad Ghazali, M. [105]	2015	Cross-sectional study	*n* = 30 University students	Kolej Jasmine, UiTM Puncak Perdana, Shah Alam, Malaysia	Adults Food Security Survey Module (AFSSM)	70% food insecurity (37% low food security, 30% very low food security)	3/8
36.	Nurulhudha, Norhasmah, Siti Nur’ Asyura, and Shamsul Azahari [106]	2020	Cross-sectional study	427 Undergraduate Public University students	Universiti Utara Malaysia, Universiti Kebangsaan Malaysia, Universiti Malaysia Pahang and Universiti Teknologi Malaysia	Adults Food Security Survey Module (AFSSM)	60.9% food insecurity (39.3% low food security, 21.6% very low food security)	7/8
37.	Roslee, R.; Lee, H.S.; Nurul Izzati, A.H.; Siti Masitah, E. [107]	2019	Cross-sectional study	*n* = 108 Low-income university students	Selangor, Malaysia	USDA Household Food Security Survey Module 2012 (Six Item Short Form)	69.4% food insecurity (50.5% low food security, 29.3% very low food security)	2/8
38.	Wan Azdie, M.A.B., Shahidah, I., Suriati, S., and Rozlin, A.R. [131]	2019	Cross-sectional study	*n* = 307 Students from six faculties of the International Islamic University Malaysia	Kuantan, Pahang	Adults Food Security Survey Module (AFSSM)	54.4% food insecurity (32.9% low food security, 21.5% very low food security)	3/8
39.	Law, L.S., Roselan, B. and Norhasmah, S. [117]	2015	In-depth interview (qualitative research)	4 informants University students	Selangor, Malaysia	-	-	5/10
Elderly population								
40.	Fadilah, M.N., Norhasmah, S., Zalilah M.S. and Zuriati I. [132]	2017	Cross-sectional study	*n* = 227 Elderly aged 60 and over	Mukim Panji, Kota Bharu District of Kelantan State	USDA Household Food Security Survey Module 2012 (Six Item Short Form)	22.9% food insecurity (15.4% low food security, 7.5% very low food security)	4/8
41.	Nurzetty Sofia, Z., Muhammmad Hazrin, H., Nur Hidayah, A., Wong, Y.H., Han, W.C., Suzana, S., Munirah, I. and Devinder Kaur, A.S. [111]	2017	Cross-sectional study	*n* = 72 Elderly aged 60 years Able to communicate total household income lower than or equal to RM3000 per month Do not suffer from terminal illness, dementia and mental illness	Urban poor residential area in Klang Valley of Malaysia (i.e., Kampung Medan, Petaling Jaya)	Radimer/Cornell Hunger and Food Insecurity Instrument	6.9% food insecurity	5/8
42.	Rohida, S.H., Suzana, S., Norhayati, I. and Hanis Mastura, Y. [112]	2017	Cross-sectional study	*n* = 289 Older adults aged 60 years and over	Older adults from FELDA Land Development Authority (FELDA)—Northern Region of Malaysia	Food Security Tool For Elderly	27.7% food insecurity 22.4% food insecurity for men 29.0% food insecurity for women	3/8
43.	Siti Farhana, M., Norhasmah, S., Zalilah, M.S., and Zuriati, I. [114]	2018	Cross-sectional study	*n* = 220 Older adults aged 60 to 87 years	Two randomly subdistricts (Petaling II and Damansara) in Petaling district	Six-item Short Form of Food Security Status	19.5% food insecurity 18.2% low food security 1.3% low food security	4/8
44.	Ruhaya et al. [113]	2020	Cross-sectional study	*n* = 3977 Older adults ≥ 60 years old	National representative sample in Malaysia	USDA Household Food Security Survey Module 2012 and USDA Household Food Security Survey Module 2012 (Six Item Short Form)	10.4% food insecurity	7/8
45.	Siti Farhana, M., Norhasmah, S., Zalilah, M.S., Zuriati, I. [114]	2020	Cross-sectional study	*n* = 220 Elderly people aged 60 years and over	Petaling district, Selangor, Malaysia	USDA Household Food Security Survey Module 2012 (Six Item Short Form)	19.5% food insecurity	6/8
Migrant worker								
46.	Chan, F.M., Faller, E.M., Lau, X.C., and Gabriel, J.S. [115]	2020	Cross-sectional study	*n* = 125 Documented migrant workers from five selected countries	Klang Valley, Selangor	Household Food Insecurity Access Scale (HFIAS)	57.6% food insecurity (24.8% mildly, 29.6% moderately, 3.2% severely)	4/8

**Table 4 nutrients-13-00945-t004:** Contributing factors, consequences, and coping strategies of food insecurity in Malaysia (40 articles).

No.	Authors	Study Design	Respondents (Target Groups)	Risk Factors for Food Insecurity	Health and Nutritional Outcomes of Food Insecurity	Categories of Coping Strategies	Coping Strategies (Top Five for Each Coping Strategy Group)
*Orang Asli*							
1.	Zalilah, M.S. and Tham, B.L. [116] (2002)	Cross-sectional study	*n* = 64 Women *Orang Asli* (Indigenous Peoples)	✓ Low income ✓ Low per-capita income	Among children ✓ Low weight-for-age and height-for-age children ✓ Poor diet quality ✓ Low intakes of calories and several nutrients (calcium and niacin)	-NA-	-NA-
2.	Nurfahilin, T. and Norhasmah, S. [92] (2005)	Cross-sectional study	*n* = 92 Women *Orang Asli*	✓ Low education level ✓ High number of children ✓ Low monthly income ✓ Large household	Among *orang Asli* women ✓ Low intake of vitamin C	Food-related coping strategies	✓ Consuming whatever food available around the house ✓ Using less expensive food ✓ Allocating money to buy staples and less preferred food ✓ Using less preferred food ✓ Cutting down the portion size or number dishes for meals
						Non-food-related coping strategies	✓ Being thrifty in using money ✓ Selling or pawning of assets ✓ Planning for expenditure ✓ Buying less expensive products or shopping at cheaper places ✓ Delaying the payment of bills ✓ Catching or fishing fish from rivers ✓ Depending on forest results
3.	Chong, S.P., Geeta, A. and Norhasmah, S. [90] (2008)	Cross-sectional study	*n* = 222 *Orang Asli* (Indigenous) Women Non-pregnant and non-lactating Non-vegetarian No chronic diseases No changes in food habits in past six months	✓ Large household ✓ Low household income ✓ Low per-capita income ✓ Low food expenditure	Among *orang Asli* women ✓ Multivariate analysis of food-insecure group was significantly associated with a higher BMI after controlling for age ✓ Child hunger had higher Malaysian HEI score for fat and sodium	-NA-	-NA-
4.	Chong, S.P., Geeta, A. and Norhasmah, S. [91] (2019)	Cross-sectional study	*n* = 222 *Orang Asli* (Indigenous) Women Non-pregnant and non-lactating Non-vegetarian No chronic diseases No changes in food habits in past six months	-NA-	Among *orang Asli* women Food security status appeared to be a predictor for diet quality	-NA-	-NA-
5.	Law, L.S, Norhasmah, S., Gan, W.Y. and Mohd Nasir, M.T. [117] (2018a)	Case study (qualitative study)	In-depth interviews 61 women (Senoi, Proto-Malay and Negrito) FGD 19 women proto Malay	-NA-	-NA-	Identified 21 coping strategies and divided into two themes	
						Food consumption (four themes)	
						Dietary changes (two subthemes)	✓ Eating plain rice with soy sauce or salt ✓ Eating less preferred food
						Diversification of food sources (nine subthemes)	✓ Finding food from surroundings ✓ Taking water from a nearby river ✓ Borrowing money to buy food ✓ Buying food on credit ✓ Buying cheaper food ✓ Buying a small amount of food ✓ Bringing children into the jungle to find food ✓ Borrowing/requesting food ✓ Borrowing motor to buy food
						Decreasing the number of people (two subthemes)	✓ Visiting friend or relative during meal ✓ Sending children to relative’s or friend’s house
						Food rationing (five subthemes)	✓ More food for children ✓ Reducing portion size ✓ Skipping meals ✓ Staying in hungry ✓ Drinking fluid when hungry
						Financial management (three themes)	
						Increasing household income (two subthemes)	✓ Taking up odd work ✓ Selling own poultry
						Reducing expenses for schooling children (three subthemes)	✓ Reducing children pocket money ✓ Ceasing to provide pocket money to children ✓ Having debt with school
						Reducing expenses on daily necessities (two subthemes)	✓ Delaying monthly electric bill payment ✓ Seldom/never buying clothes
6.	Law, L.S., Norhasmah, S., Gan, W.Y., Siti’Asyura, A., and Mohd Nasir, M.T. [118] (2018b)	Case study	*n* = 61 *Orang Asli* women Food aid-recipient household Non-pregnant and non-lactating	✓ Failure in agriculture ✓ Threat from wild animal ✓ Lack of land supply ✓ Ineffectiveness of traditional food-seeking activities ✓ Exhausting, tiring, dangerous, and time-consuming journey for food-seeking activities ✓ Depletion of natural commodities ✓ Reduced demand of natural commodities ✓ Lack of equipment ✓ Weather ✓ Rainy season ✓ Dry season ✓ Water issues ✓ Continuity of water ✓ Cleanliness of water	-NA-	-NA-	-NA-
Low-income households/welfare-recipient households/mothers							
7.	Zalilah, M.S. and Ang, M. [119] (2001)	Cross-sectional study	*n* = 137 Women (information on household food insecurity) and child (weight and height)	✓ Larger household ✓ Lower education level of mothers ✓ Lower education level of father and mother	-NA-	-NA-	-NA-
8.	Zalilah, M.S. and Khor, G.L. [100] (2004)	Cross-sectional study	*n* = 200 Women of poor households in a rural community in Malaysia	✓ Lower years of education ✓ Lower household income ✓ Low per-capita income ✓ More children ✓ Mothers as housewives	Among women ✓ Overweight ✓ Abdominal obesity ✓ At-risk waist circumference	-NA-	-NA-
9.	Zalilah, M.S. and Khor, G.L. [101] (2005)	Cross-sectional study	*n* = 200 Women of poor households in a rural community in Malaysia	✓ Lower years of education ✓ Lower household income ✓ Low per-capita income ✓ More children ✓ Mothers as housewives	Among women ✓ Overweight and obesity ✓ (more time in domestic and leisure activities) ✓ Overweight and abdominal adiposity (associated with housewives, more children, larger household, food insecurity, shorter time spent in economic activities, longer time spent in leisure activities and lower food variety score) ✓ Women from food-insecure households were significantly more likely to have at-risk WC, after controlling other factors	-NA-	-NA-
10.	Zalilah, M.S. and Khor, G.L. [102] (2008)	Cross-sectional study	*n* = 200 Women of poor households in a rural community in Malaysia	✓ Living below the poverty line ✓ No land/asset ✓ Larger household ✓ More children ✓ More school-going children ✓ Mothers as house wives		Food-related coping strategies	✓ Reduce intake of foods taken outside home ✓ Cook whatever food is available at home ✓ Reduce amount of food cooked for meals ✓ Borrow money to buy food ✓ Reduce amount of food intake
						Income/expenditure-related coping strategies	✓ Reduce daily/monthly spending ✓ Use savings ✓ Borrow money ✓ Sell valuable materials (jewellery, land, etc.) ✓ Have a second job
11.	Norhasmah, S., Zalilah, M.S., Mirnalini, K., Mohd Nasir, M.T and Asnarulkhadi, A.S. [88] (2011)	Cross-sectional study	*n* = 301 Women Non-pregnant and non-lactating, living in rural and urban areas	✓ Larger household ✓ Larger number of children ✓ Larger number of children attending school ✓ Low household income ✓ Lower per-capita income	Among women ✓ Low energy intake ✓ Low fat intake ✓ Low percentage of energy from fat ✓ Low number of servings of meat fish, or poultry and legumes ✓ Low diet diversity (in rural area)	-NA-	-NA-
12.	Norhasmah, S., Zalilah, M.S., and Rohana, A.J. [99] (2012a)	Cross-sectional study	*n* = 301 Women Non-pregnant and non-lactating, living in rural and urban areas	✓ Larger household ✓ Larger number of children ✓ Larger number of children attending school ✓ Low household income ✓ Lower per-capita income	Among women ✓ Low energy intake ✓ Low fat intake ✓ Low percentage of energy from fat ✓ Low number of servings of meat, fish, poultry and legumes Low diet diversity (in rural area)	-NA-	-NA-
13.	Mohamadpour, M., Zalilah, M.S., and Keysami, M.A. [120] (2012)	Cross-sectional study	*n* = 169 Palm plantation women Non-pregnant and non-lactating	✓ Low education ✓ Larger number of children ✓ Low household income ✓ Low per-capita income	Among women ✓ Low diet diversity ✓ Low intake of vitamin C ✓ Low serving size of protein group ✓ Diet diversity score significant protective factor against health risks even after adjusting for other variables	-NA-	-NA-
14.	Nik Aida Adibah, N.A.A. and Norhasmah, S. [109] (2013)	Cross-sectional study	*n* = 80 Women	✓ Poverty ✓ Low household income ✓ Higher percentage of household income on food expenditure	-NA-	Food-related coping strategies	✓ Using less expensive food ✓ Favoring certain household members over others ✓ Cutting down the portion or size or the number of dishes for meals ✓ Reducing the number of meals eaten in a day ✓ Consuming whatever food available around the house
						Non-food-related coping strategies	✓ Being thrifty in using money ✓ Planning for expenditures ✓ Buying less expensive products or shopping at cheaper places ✓ Buying new clothes for children but not for mothers ✓ Buying less expensive clothes or buying clothes on credit
15.	Zalilah, M.S., Norhasmah, S., Rohana, A.J., Wong, C.Y, Yong, H.Y., Mohd Nasir, M.T., Mirnalini, K., and Khor, G.L. [121] (2014)	Cross-sectional study	*n* = 625 Women Low-income communities	✓ Indian women ✓ Live in rural areas (palm plantations or villages) ✓ Have only primary education ✓ Low household income ✓ Low per-capita income	Among women ✓ Low diet diversity ✓ After adjusting for demographic and socioeconomic covariates, women with food insecurity had lower risk of: Abdominal obesity Elevated plasma glucose Cholesterol LDL cholesterol Metabolic syndrome	Food-related coping strategies (four themes)	
						Food stretching (two subthemes)	✓ Using less expensive food ✓ Using less preferred food
						Food rationing (five subthemes)	✓ Consuming whatever food available around the house ✓ Purchasing food on credit ✓ Borrowing money from employer/friends/neighbors/siblings to buy food ✓ Receiving food assistance from the agencies/neighbors/siblings/individuals/employer ✓ Sending children to eat with mothers/siblings/neighbors houses
						Food seeking (two subthemes)	✓ Consuming whatever food available around the house ✓ Purchasing food on credit
						Food anxiety (one sutheme)	✓ Allocating money to buy staples and less preferred food
						Non-food-related coping strategies (four themes)	
						Cloth purchasing behaviours (three suthemes)	✓ Buying new clothes for children but not for mothers ✓ Buying less expensive clothes or buying clothes on credit ✓ Receiving clothes from individuals or agencies
						Reduce school-going children’s expenditure (two subthemes)	✓ Children not taking money to school ✓ Decreasing children’s school pocket money
						Delay the payment of bills (three suthemes)	✓ Delaying the payment of bills ✓ Delaying the payment of bills until they received warning letters or the supplies were terminated ✓ Delaying the payment of rent
						Adjust lifestyle (8 subthemes)	✓ Being thrifty in using money ✓ Plan for expenditure ✓ Buying less expensive products or shopping at cheaper places ✓ Not attending or giving gifts during parties or festivals ✓ Requesting money from relatives or friends ✓ Increase cash and income ✓ Engaging in odd jobs ✓ Selling or pawning assets (jewelry)
16.	Cooper, E.E. [125] (2013)	Cross-sectional study	*n* = 70 Households children (*n* = 104) aged between 6 months and 6 years old Malay fishing villages	-NA-	-NA-	Food-related coping strategies	Unwanted food consumption (e.g., *uncooked blocks of ramen, diluted rice porridge and scavenged cassava leaves*) (e.g., *‘If we eat, we eat. If not, there is nothing to be done. You keep quiet’*).
						Women’s alternative income strategies	✓ Collect jungle produce/wild foods ✓ Cultivate a home garden ✓ Make ocean-derived products for sale (e.g., dried, salted fish or handicrafts) ✓ Prepare food items for sale (e.g., local cakes or satay) ✓ With licensed food stall Collect recyclables ✓ Sell items from home (e.g., grocery items, clothing, or cloth) ✓ Catch crab or fish ✓ Sew
17.	Alam, M.D., Siwar, C., Wahid, A.N.M. and Abdul Talib, B. [122] (2016)	Cross-sectional study	*n* = 460 Households from urban and rural areas East Coast Economic Region (ECER) (e-kasih database)	✓ Poverty status ✓ Marginally non-poor group significantly differ in the level of household food security from other groups (*hard-core poor, poor and recently marginally non-poor*)	-NA-	-NA-	-NA-
18.	Yong, P.P. and Norhasmah, S. [110] (2016)	Cross-sectional study	*n* = 109 Chinese women aged between 30 and 55 years old	✓ Younger age ✓ Large household ✓ Lower household income ✓ Higher number of children ✓ Higher number of schooling children ✓ Higher number of disabled family members ✓ Lower education level ✓ Lower spouse education	Among women ✓ Higher BMI ✓ At risk abdominal obesity	Food-related coping strategies	✓ Using less expensive food ✓ Using less preferred food ✓ Favouring certain household members over others ✓ Consuming whatever food available around the house ✓ Reducing the number of meals eaten in a day
						Non-food-related coping strategies	✓ Buying less expensive products or shopping at cheaper place ✓ Being thrifty ✓ Planning for expenditures ✓ Delaying the payment of the bills Delaying the payment of the rent
19.	Roselawati, M.Y., Wan Azdie, M.A.B., Aflah, A., Jamalludin, R., and Zalilah, M.S. [126] (2017)	Cross-sectional study	*n* = 128 Malay women with their children Aged 21 to 59 years olds	✓ Large household ✓ Lower income ✓ Low per-capita income ✓ Living in urban area	Among children The proportion of overweight/obesity among children from food secure households was lower (48.3%) than from food-insecure households were (60.6%) However, no significant different between childhood overweight/obesity and food security status	-NA-	-NA-
20.	Norhasmah, S., Zalilah, M.S, Mirnalini, K., Mohd Nasir, M.T and Asnarulkhadi, A.S. [123] (2012b)	Cross-sectional study	*n* = 103 Women Non-pregnant and non-lactating welfare-recipient households	✓ Presence of children below 7 years old ✓ More school-going children ✓ Disabled member(s) in the households ✓ Income reliance on financial assistance ✓ Low per-capita income	-NA-	-NA-	-NA-
21.	Siti Marhana, A.B. and Norhasmah, S. [124] (2012)	Cross-sectional Study	*n* = 80 Women and men Zakat recipients	✓ Poverty ✓ Low food expenditure compared to average Malaysian food expenditure	-NA-	Food-related coping strategies	✓ Using less expensive food ✓ Allocating money to buy staples and less preferred food ✓ Cutting down the portion size or number of dishes in a meal ✓ Reducing the number of meals eaten in a day ✓ Using less preferred food
						Non-food-related coping strategies	✓ Being thrifty in using money ✓ Planning for expenditure ✓ Delaying the payment of bills ✓ Decreasing pocket money given to school-going children ✓ Delaying the payment of bills until the received warning letters or the supplies were terminated
22.	Ihab, A.N., Rohana, A.J., Wan Manan, W.M., Wan Suriati, W.N., Zalilah, M.S. and Mohamed Rusli, A. [103](2012b)	Cross-sectional study	*n* = 223 Women Non- pregnant Non-lactating welfare-recipient households	✓ Less food group expenditure ✓ Fruits ✓ Animal-based food ✓ Milk and dairy products ✓ Decrease in food expenditure by one Malaysian Ringgit is associated with 1% increase in the odds ratio of food insecurity	-NA-	-NA-	-NA-
23.	Ihab, A.N., Rohana, A.J., Wan Manan, W.M., Wan Suriati, W.N., Zalilah, M.S., and Mohamed Rusli, A. [95] (2013)	Cross-sectional study	*n* = 223 Women Non-pregnant and non-lactating welfare-recipient households	✓ Larger household size ✓ Low total monthly income ✓ Low per-capita income ✓ Higher percentage of household income on food expenditure	Among women ✓ No significant association between food insecurity with BMI, and waist-circumference	-NA-	-NA-
24.	Ihab, A.N., Rohana, A.J., Wan Manan, W.M., Wan Suriati, W.N., Zalilah, M.S., and Mohamed Rusli, A. [54] (2012a)	Cross-sectional study	*n* = 223 Women Non-pregnant and non-lactating Welfare-recipient households	✓ Low education of the mother ✓ Larger household ✓ Larger number of children ✓ Larger number of children going to school ✓ Low monthly household income ✓ Low per-capita incomel ✓ Lower number of household members contributing to income ✓ Lower food expenditure	-NA-	-NA-	-NA-
25.	Noratikah, Norhasmah and Siti Farhana [98] (2019)	Cross-sectional study	*n* = 129 Mothers aged 20 to 59 years	✓ Low household income	Among women ✓ Depression ✓ Anxiety ✓ Stress	-NA-	-NA-
26.	Mohamad Hasnah, A., Rusidah, S., Ruhaya, S., Nur Liana, A.M., Ahmad Ali, Z., Wan Azdie, A.B., and Tahir, A. [96] (2020)	Cross-sectional study	*n* = 2962 Adults	✓ Race ✓ Education level ✓ Household income			
27.	Susanti, A., Norhasmah, S., Fadilah, M.N., and Siti Farhana, M. [127] (2019)	Cross-sectional study	n-160 Households that comprised pairs of mothers and children aged 13–17 years	✓ Number of school-going siblings ✓ Occupation status of mother ✓ Occupation status of father ✓ Household income ✓ House ownership status	Among adolescents ✓ Significant differences between food-secure and food-insecure adolescents in: Health-related quality of life BMI	-NA-	-NA-
28.	Ihab, A.N., Rohana, A.J., Wan Manan, W.M., Wan Suriati, W.N., Zalilah, M.S., and Mohamed Rusli, A. [94] (2012c)	Cross-sectional study	*n* = 223 Women Non-pregnant and non-lactating welfare-recipient households	Among women ✓ Food insecurity remains a significant independent predictor of responses for each SF-36 scale (Health-related quality of life) components	-NA-	-NA-	-NA-
University students							
29.	Norhasmah, S., Zuroni, M.J. and Siti Marhana, A.B. [128] (2013)	Cross-sectional study	*n* = 484 Undergraduate Public University students	✓ Inadequacy of student loan ✓ Increase in the cost of living ✓ Rise in tuition compulsory fees	-NA-	Food-related coping strategies	✓ Using less expensive food ✓ Cutting the portion size or number of dishes for meals ✓ Reducing the number of meals eaten in a day ✓ Cooking alone in college/rented house ✓ Sharing food with friends
						Non-food-related coping strategies	✓ Being thrifty in using money ✓ Reducing personal expenditures ✓ Planning for expenditure ✓ Requesting money from relative/friends ✓ Engaging with odd jobs
30.	Law, L.S. Roselan, B. and Norhasmah, S. [117] (2015)	Inidepth interview (qualitative research)	4 informants University students	✓ Sudden closure of cafeterias ✓ Inadequacy of loan/scholarship ✓ Lack of personal transportation ✓ Low quality of food ✓ Time constraints	Among university students ✓ Anxiety ✓ Lack of energy ✓ Inability to focus during class ✓ Falling ill	Food-related coping strategies	✓ Storing food stuff ✓ Reducing meal portions ✓ Skipping meals ✓ Requesting outside food from friends ✓ Purchasing food outside campus ✓ Forcing down dissatisfying cafeteria and mini mart meal due to hygiene issues
31.	Siti Marhana, A.B., Norhasmah, S. and Husniyah, A.R. [129] (2014)	Cross-sectional study	484 Public Undergraduate Public University students	✓ Higher percentage of household income on food expenditure (37.2%).	-NA-	Food coping strategies	✓ Using less expensive food ✓ Cutting the portion size or number of dishes for meals ✓ Reducing the number of meals eaten in a day ✓ Cooking alone in college/rented house ✓ Sharing food with friends
						Non-food-related coping strategies	✓ Being thrifty in using money ✓ Reducing personal expenditures ✓ Planning for expenditure ✓ Requesting money from relative/friends ✓ Engaging with odd jobs
32.	Nurulhudha, Norhasmah, Siti Nur’ Asyura, and Shamsul Azahari [106] (2020)	Cross-sectional study	427 Undergraduate Public University students	✓ Financial problem	-NA-	-NA-	-NA-
33.	Wan Azdie, M.A.B., Shahidah, I., Suriati, S., and Rozlin, A.R. [131] (2019)	Cross-sectional study	*n* = 307 Students from six faculties of the International Islamic University Malaysia	✓ Time constraints ✓ Spending on books ✓ Miscellaneous items ✓ Parents’ income ✓ Scholarship type	-NA-	-NA-	-NA-
34.	Nur Atiqah, A., Norazmir, M.N, Khairil Anuar, M.I., Mohd Fahmi, M. and Norazlanshah, H. [130] (2015)	Cross-sectional study	124 university students (18 to 25) years old		Among university students ✓ High Fat Mass Index ✓ High triglyceride levels among male in food insecure-group	s-NA-	-NA-
Elderly population							
35.	Fadilah, M.N, Norhasmah, S., Zalilah, M.S. and Zuriati, I. [132] (2017)	Cross-sectional study	*n* = 227 Elderly aged 60 and over	✓ Low personal income ✓ Larger household ✓ Widow/widower marital status ✓ Lower educational level ✓ Older age	-NA-	-NA-	-NA-
36.	Siti Farhana, M., Norhasmah, S., Zalilah, M.S., and Zuriati, I. [108] (2018)	Cross-sectional study	*n* = 220 Older adults aged 60 to 87 years	Univariate analysis: Significant associations between food insecurity with: ✓ Marital status ✓ Education level ✓ Occupation status ✓ Monthly income ✓ Oral health status Binary logistics: Significant likelihood to develop household food insecurity: ✓ Low monthly income ✓ Poor oral health status ✓ Living in rented accommodation	-NA-	-NA-	-NA-
37.	Ruhaya et al. [113] (2020)	Cross-sectional study	*n* = 3977 Older adults ≥ 60 years old	✓ From rural areas ✓ With primary and secondary education ✓ Income less than RM 2000 ✓ At risk of malnutrition ✓ Lack of social support	-NA-	-NA-	-NA-
38.	Nurzetty Sofia Z, Muhammmad Hazrin H, Nur Hidayah A., Wong Y. H., Han W. C., Suzana S., Munirah I. and Devinder Kaur A. S. [111] (2017)	Cross-sectional study	*n* = 72 Elderly aged 60 years Able to communicate Household income ≤ RM3000 per month No terminal illness, dementia and mental illness	-NA-	Among elderly population ✓ Bivariate analysis Significant correlation between food insecurity, low height of respondents and low fat intake	-NA-	-NA-
39.	Rohida, S.H., Suzana, S., Norhayati, I. and Hanis Mastura, Y. [112] (2017)	Cross-sectional study	*n* = 289 Older adults aged 60 years and over from FELDA Land Development Authority (FELDA)—Northern Region of Malaysia	-NA-	Among elderly population ✓ Significant higher intake of fat, oil, sugar, and salt among food-insecure elderly	-NA-	-NA-
40.	Siti Farhana, M., Norhasmah, S., Zalilah, M.S., Zuriati, I. [114] (2020)	Cross-sectional study	*n* = 220 Elderly people aged 60 years and over	-NA-	Among elderly population ✓ Significant correlation between food insecurity and depression	-NA-	-NA-

## Data Availability

Not applicable.

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
