# Peer review of "A Food Insecurity Systematic Review: Experience from Malaysia"

_nutrients, 2021, doi:10.3390/nu13030945_

Round 1
Reviewer 1 Report
In this review of food insecurity in Malaysia, the authors conduct a systematic literature review and report of the prevalence of household food insecurity, risk factors for food insecurity, coping strategies employed by different groups, and the consequences of food insecurity.
Literature review: The literature review jumps around touching on studies from around the world without going into much depth about any singular aspect of food insecurity. At the end, the authors mention Malaysia, without providing much context for studying food insecurity in Malaysia, why it should be studied, and how it can contribute to increasing food security.
The authors would be wise to focus the literature review more in answering one or two major research questions and setting up Malaysia as a case study in answering those research questions. As is, only one paragraph is dedicated to describing the country. The authors state the "poverty rate in Malaysia is low, at less than one percent". How is poverty rate measured? If the poverty rate is so low, why do the 33 studies find high levels of food insecurity, given that lack of money was a major reason for individual food insecurity? I have a hard time wrapping my head around these seemingly conflicting statistics. The authors also provide a range for gross national income. Why is there a range? Why not just report the median individual income for the last year collected (2020 or 2019)? More description of Malaysia's economic and demographic background is needed for readers to better situate the study's results. For instance, while the authors claim only 1% of population is impoverished they claim that 31.2% of Orang Asli population live below the poverty line. What % of the Malaysian population are Orang Asli? Are their other ethnic or indiginous groups that have high poverty rates? Knowing more about the economic and demographic/ethnic history of Malaysia is important for the reader.
The authors provide a paragraph about the limitations of the study. This should be expanded quite a bit to describe the limitations of the 33 studies of food security in Malaysia. For example, they seem to be quite small. How strong is the methodology for sampling? How generalizable are the results? Are there any populations not studied that should be studied? Are there aspects of food security/insecurity that are not being researched but ought to?
Tables/Charts: Chart 1 lacks spacing at the bottom. The arrow and diamond that describe total included articles overlap the fourth stage.
Most of the tables are descriptions of the 33 articles used for the review. They seem to take up a lot of space in the article, and I don't know how useful they all are. I tended to gloss over them. Maybe, some of the bigger tables (tables 5 and 6) could be an appendix.
The study's major purpose and main takeaways are not clear. While the paper is a meta analysis, the study should still seek to answer one or more major social research questions and describe why these questions are important and how the study can help government officials, public policy, activists, other researchers, etc. to reduce food insecurity.
Reviewer 2 Report
The paper is well-written and easy to follow. I have just minor suggestions.
line 130-131: It is not clear because "the period of publication was from August 1957".
Chart 1.0: Article Selection Processes: the last block is superimposed on the penultimate.
Paragraph 3.1. Prevalence of Food Insecurity: This paragraph could be improved in clarity insert in the first sentence the names of the identified sub-populations.
Reviewer 3 Report
The manuscript submitted by Sulaiman et al. presents an in-depth analysis of Food Insecurity in Malaysia. From my point of view, the study is quite well designed and presented, providing important results in this topic.
I only suggest one minor change:
The authors should state in the title that this is a systematic review.
Author Response
Please see my attachment

Round 2
Reviewer 1 Report
Review of "A Food Insecurity Systematic Review: Experience from Malaysia" After reading the revised version of the article, many of my comments/concerns have been addressed. This version is now stronger as it has clearly defined research questions and the discussion section has been strengthened. The tables are now more succinct and some explanation of the poverty /ethnicity statistics now exist in the literature review. The authors chose not to focus their literature review and instead stuck with a broad, yet shallow, literature review. That is their choice, but I don't believe it provides the type of intellectual depth needed to understand the very specific situation of food insecurity in Malaysia. I still would like to see the literature review to be focused on Malaysia. By doing this, the results of the systematic review are placed into context that readers who know little about Malaysia can understand. However, the addition of the demographic ethnic background statistics, poverty line, and median income statistics helps and I'm glad they added them.Author Response
Thank you for your criticism and correction. We have made corresponding modifications in the article